# Multimodal biomarker based on temporal complexity of eye movements and pupil diameter in attention-deficit/hyperactivity disorder

**Ayumu Ueno**[1]⊛*, **Sou Nobukawa**[1,2,3]⊛*, **Aya Shirama**[2,4]

**1** Graduate School of Information and Computer Science, Chiba Institute of Technology, Narashino, Japan, **2** Department of Preventive Intervention for Psychiatric Disorders, National Institute of Mental Health, National Center of Neurology and Psychiatry, Tokyo, Japan, **3** Research Center for Mathematical Engineering, Chiba Institute of Technology, Narashino, Japan, **4** Graduate School of Data Science, Nagoya City University, Aichi, Japan

⊛ These authors contributed equally to this work.
* s2131021kw@s.chibakoudai.jp (AU); nobukawa@it-chiba.jp (SN)

**Data availability statement:** All relevant data are within the paper and its Supporting information files. The de-identified,

## Abstract

Attention-deficit/hyperactivity disorder (ADHD) is one of the most prevalent neurodevelopmental disorders without established biomarkers. Pupil diameter, regulated by the locus coeruleus-norepinephrine (LC-NE) system, and eye movements controlled by diverse brain regions exhibit specific patterns in patients with ADHD due to abnormal activity in these brain regions. Previous studies reported that patients with ADHD have larger pupil diameters and reduced temporal complexity. However, although the temporal complexity of eye movements has been associated with major movement disorders in previous studies, its relevance in conditions without primary movement disorders, such as ADHD, remains unclear. In this context, we hypothesized that the temporal complexity of eye movements would provide a more comprehensive understanding of eye movements in ADHD. This study aimed to analyze the temporal complexity of eye movements in patients with ADHD using multiscale entropy analysis and examine its diagnostic utility alongside pupil size. The results showed that patients with ADHD had lower temporal complexity in their eye movements and larger pupil diameters. Moreover, combining these features enhanced the accuracy of ADHD diagnosis. These findings support the potential of a multimodal approach for diagnosing adult ADHD, potentially improving clinical diagnostic accuracy.

## Introduction

Attention deficit hyperactivity disorder (ADHD) is a neurodevelopmental disorder characterized by inattention, impulsivity, and hyperactivity that occurs primarily in childhood [1]. The disorder often persists into adulthood [2], and according to a World Health Organization (WHO) survey, the average prevalence of ADHD in adults is approximately 2.8% [3]. Precise diagnosis and subsequent intervention are critical for adults with ADHD because they

participant-level minimal dataset required to reproduce all results is provided as S1 Data (CSV; S1_data.csv), including group labels, age, sex, medication status, pupil measures, multiscale entropy features, exclusion ratios, and measurement metadata.

**Funding:** This study was supported by JSPS KAKENHI through a Grant-in-Aid for Scientific Research (C) (Grant No. JP23K03024 to AS), a Grant-in-Aid for Scientific Research (B) (Grant No. JP25K03198 to SN), and a Grant-in-Aid for Transformative Research Areas (A) (Grant No. JP20H05921 and Grant No. JP25H02626 to SN). This work was partially supported by Project JPNP14004, commissioned by the New Energy and Industrial Technology Development Organization (NEDO). The funders had no role in study design, data collection and analysis, decision to publish, or preparation of the manuscript.

**Competing interests:** We have applied for a patent related to the proposed methods (Japanese Patent Application No.2024-225037). The authors have declared that no other competing interests exist.

often suffer from psychological and social dysfunction without proper diagnosis and treatment [4,5]. Therefore, it is expected that biomarkers will be established to achieve objective and quantitative assessments to support precise diagnosis [6,7], in addition to conventional diagnosis based mainly on medical interviews [8].

Various biological signals have been investigated as potential ADHD biomarkers, including electroencephalography (EEG), functional magnetic resonance imaging (fMRI), near-infrared spectroscopy, and eye tracking (reviewed in [9]) [10–14]. Among these measurements, eye tracking is particularly effective in clinical applications because of its cost-effectiveness, time efficiency, wide availability, and noninvasive nature [15]. One type of time series captured by eye tracking is the pupil diameter. Pupil diameter is strongly influenced by the locus coeruleus (LC) [16], which is a common source of both the sympathetic pathway to the pupil dilator muscle and the parasympathetic pathway to the pupil sphincter muscle [17,18]. Additionally, the LC is the major source of norepinephrine in the forebrain, with most cortical and subcortical areas receiving dense LC-NE axonal innervation [19]. In ADHD, structural and functional abnormalities have been reported in the LC-NE brain regions (reviewed in [20]). Consequently, this overlap of brain regions causes abnormal pupillary behavior in ADHD [14,21,22]. In addition to the pupil diameter, eye movement is another time series captured by eye tracking. The neural network controlling eye movements is a complex system consisting of diverse brain regions, including the cortical, subcortical, and cerebellar regions, which play roles in sensory, motor, attention, and executive functions (reviewed in [23]). In ADHD, the brain regions that control eye movement also overlap with structural and functional abnormalities (reviewed in [20]), which lead to abnormal ADHD eye movement [24]. Utilizing these characteristics involving different regional neural activities through eye trackers could significantly contribute to a comprehensive understanding of ADHD pathology.

The analysis of pupil diameter and eye movement often involves measuring the temporal average of the pupil diameter, emergence frequency and duration of fixation, amplitude and direction of saccades, and direction and velocity of smooth pursuit (reviewed in [25]). In addition to these conventional metrics, temporal complexity—defined as the degree of irregularity or unpredictability of time-series patterns—is quantified using entropy-based indices, such as sample entropy and fuzzy entropy (FuzzyEn); higher values indicate greater irregularity, and this approach has recently been applied to a variety of physiological signals [14,22,26–30]. Temporal complexity in neural activity is widely used and has been reported to represent the degree of interaction of neural activity between regions, and to play an important role in precisely controlling hierarchical neural activities [31–34]. Additionally, the temporal complexity of pupil diameter has been found to reflect cognitive processes and neural activity in both healthy and pathological states [14,22,26–28]. Particularly in ADHD, the temporal complexity of pupil diameter is important for capturing the pathophysiology of ADHD, and its relationship with the LC-NE system has been reported [14,22]. Furthermore, a study on patients with cerebellar ataxia showed significant differences in the temporal complexity of eye movements between patients with and without dysmetria (a condition characterized by impaired movement coordination, often resulting in an undershoot or overshoot of the intended target) [30]. However, even in psychiatric disorders without movement disorders as the primary complaint, it remains unclear whether alterations in the temporal complexity of eye movements occur. Additionally, evaluating the temporal dependence of temporal complexity across multiple timescales may capture characteristics at specific timescales which are related to the dysfunction of fixational eye movement control, such as tremors, microsaccades, and drift [25].

We hypothesized that the timescale dependence of temporal complexity in eye movements would provide a more comprehensive understanding of eye movements in ADHD. Moreover, a classification method that combines these characteristics with pupil diameter may contribute to the establishment of biomarkers for ADHD. This study aimed to assess pupil diameter and eye movement during a resting fixation task in adults with typical development (TD) and ADHD. Specifically, a temporal average was calculated to evaluate the mean pupil diameter, and multiscale entropy (MSE) analysis based on FuzzyEn was performed to evaluate the profile of temporal complexity across multiple time scales of eye movement. The ability to classify TD and ADHD was assessed based on these characteristics.

## Results

### Analysis of eye movement dynamics

An overview of the procedure used in this study to analyze eye movement and pupil diameter is shown in Fig 1A. The participants included 20 adults with TD and 16 adults with ADHD. Additionally, a drug-naïve ADHD group of 11 individuals was created by removing the effects of ADHD medications. The eye movements and pupil diameters of the participants during the fixational eye movement task were recorded using an eye tracker and analyzed after each preprocessing step. For the analysis of eye movement, MSE analysis was used to evaluate the time-scale dependence of the temporal complexity of the time-series [35]. In the MSE analysis, the eye movement time series was coarse-grained at each temporal scale; these temporal complexities were quantified using FuzzyEn. Additionally, the analysis of mean pupil diameter was conducted using the temporal mean pupil diameter. Subsequently, the ability to classify TD and ADHD was evaluated using Lasso logistic regression based on the results of MSE for eye movement and pupil diameter.

First, we analyzed iterative amplitude-adjusted Fourier transform (IAAFT) surrogate data to investigate whether nonlinear dynamics reflecting intrinsic neural activity are present in the temporal complexity of eye movement across multiple timescales. This analysis was performed by comparing the MSE profile of the eye-movement time series with that of the corresponding IAAFT surrogates in each group (TD, ADHD, and drug-naïve ADHD). Fig 1B shows the comparison between FuzzyEn of the original eye movements and the average FuzzyEn of 10 IAAFT surrogates generated from different random seeds for each temporal scale [1–30 (0.003–0.1 s)]. The FuzzyEn values at each temporal scale had a skewed distribution and were therefore log-transformed to approximate a normal distribution (see S1 Fig for representative probability-density plots of the scale-averaged features before and after transformation) [36,37]. For horizontal and vertical eye movements in all groups (TD, ADHD, and drug-naïve ADHD), the FuzzyEn values of the IAAFT surrogates were significantly higher than those of the original time series on a temporal scale [$\leq 30$ ($\leq 0.1$ s)]. Thus, these results indicated that the time-series patterns of horizontal and vertical eye movements reflected nonlinear dynamics, as they cannot be explained by the null hypothesis of a linear stochastic process.

Second, as shown in Table 1, the MSE profiles of these eye movements were evaluated using repeated-measures analysis of covariance (ANCOVA) with group (TD vs ADHD, TD vs drug-naïve ADHD) as a between-subjects factor, temporal scale [1–30 (0.003–0.1 s)] as a within-subjects factor, and age as a covariate. Comparisons between the TD and ADHD groups revealed a significant main effect of group and interaction on the group × temporal scale for vertical eye movement. Furthermore, comparisons between the TD and drug-naïve ADHD groups revealed a significant main effect of group and interaction on the group × temporal scale for horizontal and vertical eye movements. The results of the post

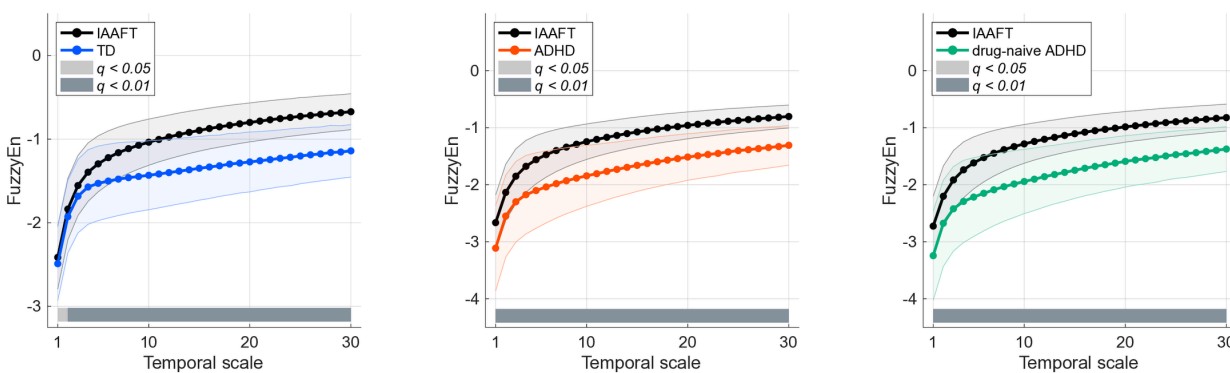

**Fig 1. Overview of analysis procedures (A) and nonlinear dynamics of eye movements (B).** (A) Overview of the procedure for analysis of eye movement and pupil diameter (left part). An image of the fixation task (upper right part) and the composition of the participants of groups (TD, ADHD, drug-naïve ADHD) (lower left part) are shown. (B) Comparison between the MSE profile of the original eye movement time-series and that of the corresponding IAAFT surrogate. In all groups, the time-series patterns of horizontal (upper part) and vertical (lower part) eye movements reflected nonlinear dynamics in the temporal scale [$\leq 30$ ($\leq 0.1$ s)]. Error bars indicate standard deviation. false discovery rate (FDR) correction criteria: $q < 0.05$ and $q < 0.01$.

**Table 1. Repeated-measures ANCOVA results for MSE profiles of eye movements across groups.**

| | Group effect | Group × Scale Factor |
|---|---|---|
| *TD vs ADHD* | | |
| Eye movement (Hor) | $F = 3.351, p = 0.076, \eta^2 = 0.092$ | $F = 0.944, p = 0.347, \eta^2 = 0.028$ |
| Eye movement (Vert) | **$F = 5.280, p = 0.028, \eta^2 = 0.138$** | **$F = 12.360, p < 0.001, \eta^2 = 0.272$** |
| *TD vs drug-naïve ADHD* | | |
| Eye movement (Hor) | **$F = 5.581, p = 0.025, \eta^2 = 0.166$** | **$F = 6.096, p = 0.016, \eta^2 = 0.179$** |
| Eye movement (Vert) | **$F = 7.804, p = 0.009, \eta^2 = 0.218$** | **$F = 11.940, p = 0.001, \eta^2 = 0.299$** |

The MSE profiles of eye movement were evaluated using repeated-measures analysis of covariance (ANCOVA) with group (TD vs ADHD, TD vs drug-naïve ADHD) as a between-subjects factor, temporal scale [1–30 (0.003–0.1 s)] as a within-subjects factor and age as a covariate. The significant group difference ($p < 0.05$) is represented by bold text.

hoc *t*-test (horizontal eye movement with group [TD vs drug-naïve ADHD] and vertical eye movement with group [TD vs ADHD and TD vs drug-naïve ADHD]), which indicated significant differences in the ANCOVA, are shown in Fig 2. The results of horizontal eye movements showed significantly lower FuzzyEn in drug-naïve ADHD in temporal scales of [≤ 30 (≤ 0.1 s)] ($q < 0.05$) and [≤ 16 (≤ 0.053 s)] ($q < 0.01$). Additionally, the results of vertical eye movements showed significantly lower FuzzyEn in ADHD in the temporal scale [≤ 12 (≤ 0.04 s)] ($q < 0.05$) and significantly lower FuzzyEn in drug-naïve ADHD in the temporal scale [≤ 18 (≤ 0.06 s)] ($q < 0.05$).

## Analysis of mean pupil diameter

The mean pupil diameter was analyzed in the groups (TD, ADHD, and drug-naïve ADHD) using the temporal mean values of pupil diameters. The pupil size was evaluated using ANCOVA with group (TD vs ADHD or drug-naïve ADHD) as a between-subjects factor and age as a covariate. In the group comparison between TD and ADHD, a significant main effect of the group on pupil size was observed ($F = 8.532, p = 0.006, \eta^2 = 0.205$). Similarly, in the group comparison between TD and drug-naïve ADHD, there was a significant

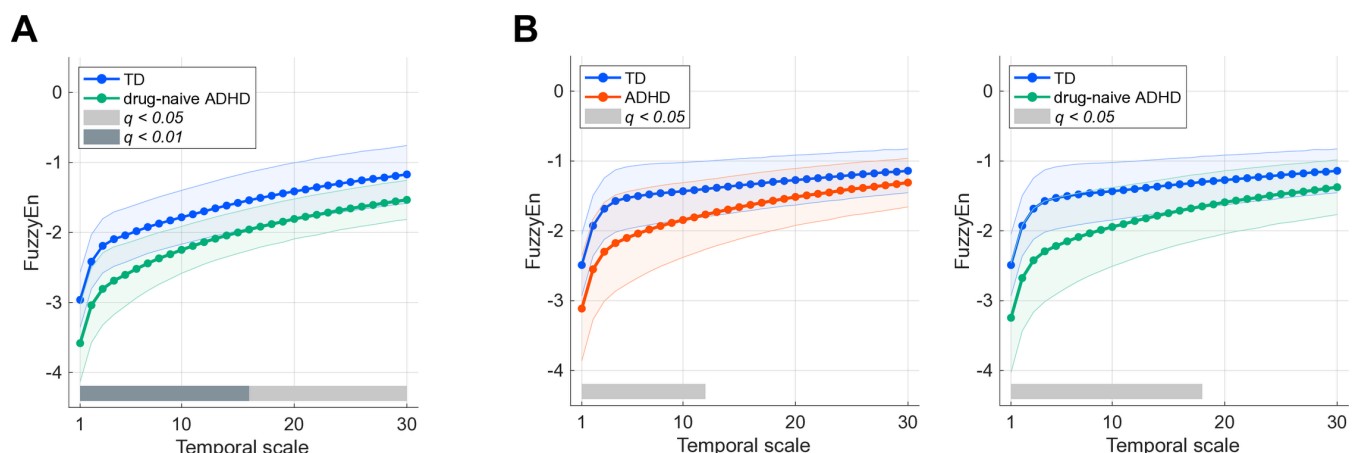

**Fig 2. Post hoc *t*-test comparisons of FuzzyEn across temporal scales in groups with significant ANCOVA results.** As the post hoc *t*-test, The average of FuzzyEn dependency on the temporal scale in the groups (TD, ADHD, and drug-naïve ADHD) were compared in the groups that were significant in the ANCOVA. (A) Comparison of horizontal eye movement in the group (TD vs drug-naïve ADHD). (B) Comparison of vertical eye movement between the groups (TD vs ADHD or drug-naïve ADHD). Error bars indicate standard deviation. FDR correction criteria: $q < 0.05$ and $q < 0.01$.

main effect of group on pupil size ($F$ = 4.339, $p$ = 0.046, $\eta^2$ = 0.134). In a post hoc $t$-test of ANCOVA, the comparison of the TD and ADHD groups showed a significantly larger pupil size in the ADHD group ($t$ = 3.477, $p$ = 0.001). Similarly, a comparison of the TD and drug-naïve ADHD groups revealed significantly larger pupil size in drug-naïve ADHD group ($t$ = 3.144, $p$ = 0.004).

## Evaluation of classification ability using characteristics of eye movement and pupil diameter

We evaluated the performance of Lasso logistic regression models in discriminating between TD and ADHD participants [38], as well as between TD and drug-naïve ADHD participants. Three features were considered: mean pupil diameter (Pupil Size), Fuzzy Entropy of horizontal eye movements (Hor FuzzyEn), and Fuzzy Entropy of vertical eye movements (Vert FuzzyEn). The Hor FuzzyEn and Vert FuzzyEn features were calculated by averaging values on a temporal scale [1–10 (0.003–0.03 s)] for each log-transformed MSE value. A preliminary analysis using Pearson's correlation revealed a high correlation between Hor FuzzyEn and Vert FuzzyEn (see S1 Table), suggesting potential information redundancy. Therefore, to identify the optimal feature set, we systematically compared the performance of classifiers constructed from all possible feature combinations. The performance of the primary models is summarized by Receiver Operating Characteristic (ROC) and precision-recall (PR) curves in Fig 3, while a complete comparison is detailed in S2 Table.

The performance of the primary classifiers was evaluated using the Area Under the ROC Curve (AUC-ROC) and the Area Under the PR Curve (AUC-PR), with the curves presented in Fig 3A and 3B. For the TD vs ADHD group classification, combining features markedly improved performance over single-feature models across both metrics (see Fig 3A). The single-feature AUC-ROC values were 0.76 (Pupil Size), 0.65 (Hor FuzzyEn), and 0.72 (Vert FuzzyEn). The models combining pupil and entropy information achieved higher values of 0.79 (Pupil Size + Hor FuzzyEn) and 0.83 (Pupil Size + Vert FuzzyEn). For a comprehensive comparison, other combinations were also evaluated (see S2 Table). The model combining only the two redundant entropy features yielded the lowest two-feature performance, with an AUC-ROC of 0.72 (Hor FuzzyEn + Vert FuzzyEn). Furthermore, while the three-feature model improved upon the single-feature models with an AUC-ROC of 0.79 (Pupil Size + Hor FuzzyEn + Vert FuzzyEn), it did not achieve the best overall performance. This trend was mirrored in the AUC-PR values. Specifically, single-feature models yielded AUC-PRs ranging from 0.59 to 0.71, whereas the models combining pupil and entropy information produced markedly superior results of 0.73 (Pupil Size + Hor FuzzyEn) and 0.79 (Pupil Size + Vert FuzzyEn). This outcome indicates that combining pupil-derived and eye-movement-derived features improved the classification performance across both metrics.

A similar analysis was conducted for the TD vs drug-naïve ADHD group classification, which yielded comparable findings (see Fig 3B). Once again, combining features improved the performance over single-feature models. The single-feature AUC-ROC values were 0.77 (Pupil Size), 0.78 (Hor FuzzyEn), and 0.75 (Vert FuzzyEn). The models combining pupil and entropy information achieved high values of 0.83 (Pupil Size + Hor FuzzyEn) and 0.82 (Pupil Size + Vert FuzzyEn). For a comprehensive comparison, the other combinations detailed in S2 Table were also considered. The model combining only the two entropy features again showed lower performance, with an AUC-ROC of 0.75 (Hor FuzzyEn + Vert FuzzyEn). Similarly, the three-feature model, with an AUC-ROC of 0.78 (Pupil Size + Hor FuzzyEn + Vert FuzzyEn), improved upon the single-feature models but did not surpass the best-performing

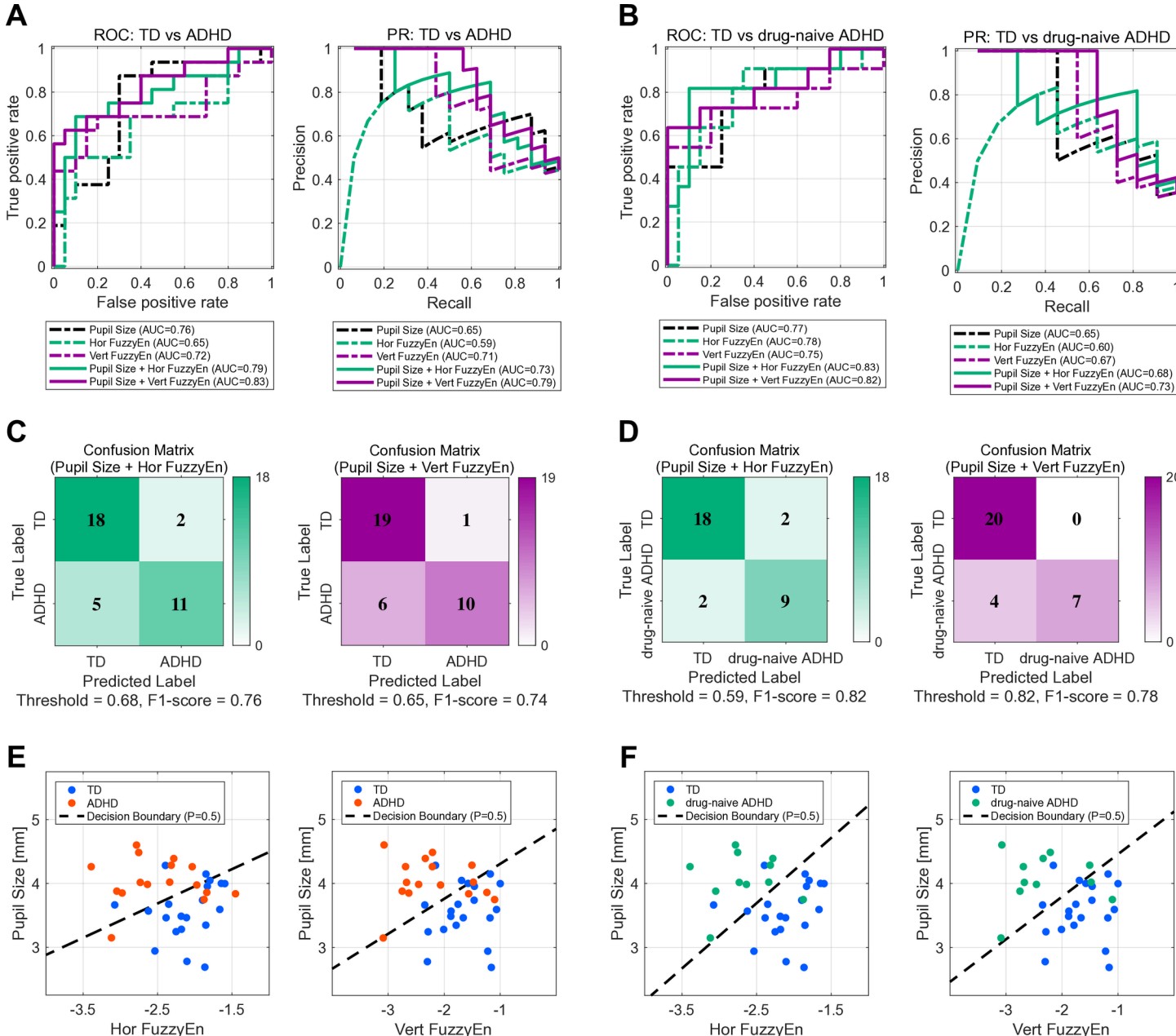

**Fig 3. Classification performance for discriminating between TD and ADHD groups using Lasso logistic regression models.** The left column (A, C, and E) shows the results for TD vs ADHD. The right column (B, D, F) shows the results for TD vs drug-naïve ADHD. (A, B) Receiver operating characteristic (ROC; left) and precision–recall (PR; right) curves comparing single-feature classifiers (Pupil Size, Hor FuzzyEn, Vert FuzzyEn) with two-feature classifiers (Pupil Size + Hor FuzzyEn; Pupil Size + Vert FuzzyEn). (C, D) Confusion matrices for the Pupil Size + Hor FuzzyEn (left) and Pupil Size + Vert FuzzyEn (right) classifiers; thresholds were chosen to maximize the F1-score on each classifier's PR curve. (E, F) Decision boundaries in the Pupil Size vs Hor FuzzyEn (left) and Pupil Size vs Vert FuzzyEn (right) feature planes.

two-feature combination model. This pattern was consistently reflected in the AUC-PR values. Specifically, single-feature models yielded AUC-PRs ranging from 0.60 to 0.67, whereas the combined models produced markedly superior results of 0.68 (Pupil Size + Hor FuzzyEn) and 0.73 (Pupil Size + Vert FuzzyEn). These results also demonstrate that

combining pupil- and eye-movement-derived features improved the classification performance across both metrics.

The practical performance of the two-feature classifiers is presented in the confusion matrices shown in Fig 3C and 3D. The classification threshold for each model was selected to maximize the F1-score on its respective PR curve (Fig 3A and 3B) [39]. For the TD vs ADHD classification, the Pupil Size + Hor FuzzyEn model achieved an F1-score of 0.76 (sensitivity: 68.8%, specificity: 90.0%), and the Pupil Size + Vert FuzzyEn model achieved an F1-score of 0.74 (sensitivity: 62.5%, specificity: 95.0%). For the TD vs drug-naïve ADHD classification, the Pupil Size + Hor FuzzyEn model achieved an F1-score of 0.82 (sensitivity: 81.8%, specificity: 90.0%), and the Pupil Size + Vert FuzzyEn model achieved an F1-score of 0.78 (sensitivity: 63.6%, specificity: 100.0%).

Finally, Fig 3E and 3F visualize the linear decision boundaries ($P = 0.5$) for the two-feature classifiers in their respective feature planes. These plots illustrate how the models combine information from both pupil size and FuzzyEn to separate the groups. The slope of the boundaries visually confirms that both features in each pair contribute to the classification decisions for both the general and drug-naïve ADHD datasets.

Additionally, we investigated how pupil size, Hor FuzzyEn, and Vert FuzzyEn are related to the severity and symptoms of ADHD. Fig 4 shows scatter plots of pupil size, Hor FuzzyEn, and Vert FuzzyEn against the ASRS score, as well as the ASRS subscores, which include the ASRS hyperactivity/impulsivity (ASRS Hyp/I) score and the ASRS inattentive (ASRS IN) score. The correlation coefficients $r$ for these relationships involving total scores and subscores exhibited no correlation in either the ADHD or drug-naïve ADHD groups.

## Discussion

This study aimed to investigate the dynamics of eye movements in individuals with ADHD. To achieve this, we evaluated the timescale dependence of temporal complexity across multiple temporal scales of eye movements. Additionally, we examined whether these characteristics, combined with an index of pupil diameter, could contribute to the development of ADHD biomarkers. Surrogate data analysis revealed that the temporal complexity of eye movements reflected nonlinear dynamics in all the groups (TD, ADHD, and drug-naïve ADHD). Furthermore, in the comparison between the TD and ADHD groups, the temporal complexity of vertical eye movements was significantly lower in the ADHD group. In the comparison between the TD and drug-naïve ADHD groups, the temporal complexity of both horizontal and vertical eye movements was significantly lower in the drug-naïve ADHD group. Consistent with previous studies, pupil size was significantly larger in both the ADHD and drug-naïve ADHD groups than in the TD group [14,21]. Moreover, a systematic comparison of classifiers revealed that the highest classification accuracy was achieved by combining pupil size and a single measure of temporal complexity of eye movements, an approach that outperformed models based on individual features. Finally, our investigation into the relationship between pupil size, temporal complexity of eye movements, and ADHD severity and symptoms revealed no significant correlations.

Understanding why nonlinear dynamics are reflected in eye movements, as shown in Fig 1B, requires consideration of the underlying neural mechanisms. These results suggest that eye movement patterns are not merely the product of linear stochastic noise but reflect specific eye-controlling mechanisms supported by intrinsic neural activity. Functional neuroimaging studies have revealed that the brain regions that control eye movement are complex systems involving diverse cortical areas (frontal eye fields, supplementary eye fields, dorsolateral prefrontal cortex, cingulate eye field, parietal eye field, precuneus, and middle

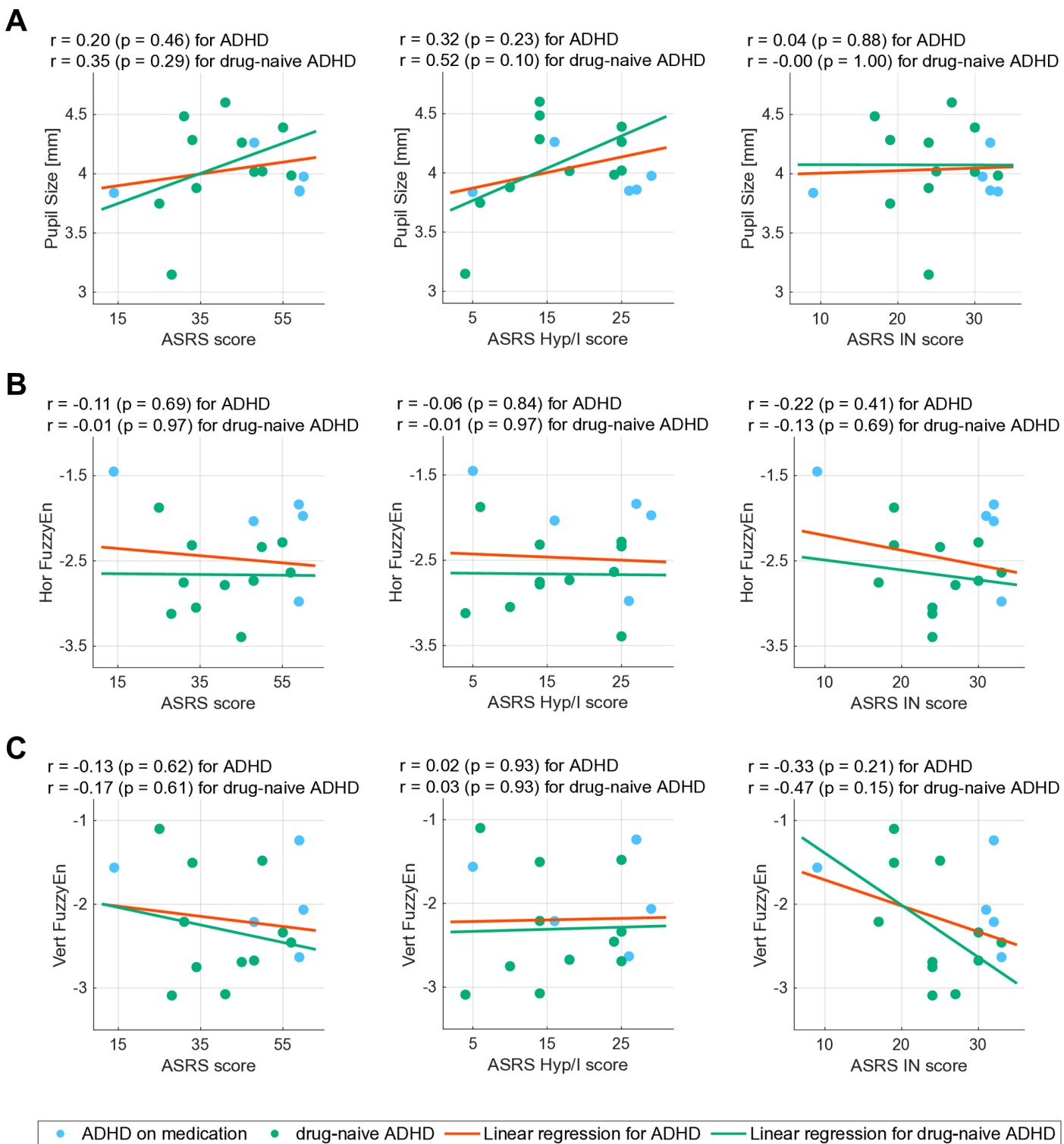

**Fig 4. Scatter plots and correlation analysis of pupil size and FuzzyEn measures against ASRS scores.** Scatter plots of pupil size, Hor FuzzyEn, and Vert FuzzyEn against scores on the Adult ADHD Self-Reporting Scale (ASRS; left). Scatter plots with ASRS subscores, which are the ASRS hyperactivity/impulsivity (ASRS Hyp/I) score (middle) and the ASRS inattentive (ASRS IN) score (right). Pearson's correlation coefficient *r* and *p*-value for pupil size, Hor FuzzyEn, and Vert FuzzyEn against the ASRS score are represented in the upper part of each figure. *r*, *p*-value, and linear regressions are calculated for the ADHD group, which includes both drug-naïve participants and those on medication, as well as for the drug-naïve ADHD group alone. Pupil size, Hor FuzzyEn, and Vert FuzzyEn were not correlated with ADHD severity or symptoms in either group. (A) Scatter plot of pupil size and each ASRS score. (B) Scatter plot of Hor FuzzyEn and each ASRS score. (C) Scatter plot of Vert FuzzyEn and each ASRS score.

temporal complex) and subcortical areas (thalamus, striatum, superior colliculus, brainstem motor nuclei, and cerebellum) (reviewed in [23,40–43]). These regions are highly interconnected and integrate diverse sensory, attentional, and motor information to achieve accurate eye movements (reviewed in [23,44]). Thus, the nonlinear dynamics of eye movements may reflect the diverse intrinsic neural activities in these brain regions.

Additionally, it is crucial to consider why the temporal complexity of eye movements in patients with ADHD was low, as shown in Fig 2. This finding suggests that eye movements in ADHD are more regular than those in the TD group. The neural mechanisms of attentional control are closely related to those of microsaccadic eye movements [45,46]. Previous studies have reported that patients with ADHD have a reduced ability to inhibit microsaccades and that the frequency of microsaccades is positively correlated with ADHD characteristics [13,47,48]. These microsaccades have been shown to be involuntary and relatively regular movements, with their direction and frequency influenced by attention and other factors [49–51]. Therefore, specific eye movement patterns in individuals with ADHD may be more regular, leading to a lower temporal complexity of eye movements. In contrast, previous studies on patients with cerebellar ataxia reported that the group with abnormal saccadic eye movements (i.e., hypometric or hypermetric saccades) exhibited higher temporal complexity in their eye movements than the group without abnormalities [30]. These abnormal saccadic eye movements are thought to result from the inhibition of smooth saccades, leading to irregular movements and, consequently, higher temporal complexity of eye movements. Therefore, changes in the temporal complexity of eye movements may reflect the characteristics of the underlying eye-controlling mechanisms, with different patterns observed in various pathologies, such as ADHD and cerebellar ataxia. Hence, analyzing the temporal complexity of eye movements is important for understanding the functional status of the brain.

Furthermore, it is important to consider why combining pupil size and the temporal complexity of eye movements provided a higher classification accuracy than using each parameter individually, as shown in Fig 3A, 3B. Previous studies have indicated LC-NE system dysfunction in individuals with ADHD [19,20], which manifests as abnormal pupil diameter responses [14,21,22]. Additionally, ADHD is associated with dysfunctions in various eye-controlling brain regions, such as the precuneus, thalamus, striatum, and cerebellum [20,23,52–54], leading to abnormal eye-movement responses [24]. The combination of physiological markers of abnormalities in these different brain regions enabled a more accurate representation of the multifaceted changes in neurological function, resulting in the relatively high classification accuracy achieved in this study. However, our findings also provide crucial insights into the limitations of this approach. As detailed in S2 Table, our comparison revealed two key limitations to simply adding features: the model combining only the two redundant entropy features was among the least effective two-feature classifiers, and the full three-feature model was also outperformed by the more parsimonious two-feature models. This outcome is likely attributable to two factors: the risk of overfitting in a limited sample size, and the information redundancy from the two highly correlated entropy features, as shown in our preliminary analysis (S1 Table). This underscores the importance of the principle of parsimony in biomarker development, suggesting that the optimal model is one that balances predictive power with simplicity by combining complementary, non-redundant information sources. Moreover, recent studies have focused on multimodal approaches that combine features such as pupil diameter, eye movement, EEG, and fMRI [55]. These studies highlighted the complementary relationship between each biological signal and reported that combining them improves the accuracy of pathological condition detection [55]. Our study results support the usefulness of these multimodal approaches in the diagnosis of ADHD and may contribute to improving diagnostic accuracy in clinical practice in the future.

This study provided valuable insights into the dynamics of eye movements in patients with ADHD and their potential as biomarkers; however, it has some limitations. First, the primary limitation of this study stems from its dataset characteristics, namely the relatively small sample size and moderate class imbalance between the groups (see Table 2). This affects our findings from two main perspectives: the generalizability of group-level differences and the reliability of the classification model. From the first perspective, although we observed significant differences in eye movement dynamics between the groups, a larger and more diverse cohort is necessary to confirm that these findings are broadly generalizable to the entire adult ADHD population and to better disentangle the influence of confounding factors, such as medication. From the second perspective, developing a diagnostic classifier on a small and imbalanced dataset presents challenges regarding model stability and the potential for overfitting. Although we implemented robust techniques designed for such conditions, including L1-regularized regression and a weighted-class approach [38,56], the reported classification performance should be considered as preliminary. Notably, however, the principal finding that a combination of pupil and eye-movement features yields superior classification performance was consistent across all evaluated models, including standard and Firth's logistic regression (S3 Table) [57]. Consequently, while the proposed model shows potential as a biomarker, its reliability and clinical utility must be established through validation using larger, independent datasets.

Second, a potential limitation concerns the possibility of head motion artifacts, particularly in the ADHD group. To address this issue, we performed several analyses. First, heatmap analysis of raw gaze distributions revealed that, although gaze points in both the ADHD and drug-naïve ADHD groups remained concentrated near the central fixation cross, their spatial dispersion was significantly larger than that of the TD group (S1 Text). This increased dispersion aligns with previous findings indicating increased fixational instability and a higher frequency of microsaccades in ADHD [47,58,59]. However, this observation alone does not fully exclude potential contamination by head motion artifacts. Previous studies in adults have established that the power of involuntary head motion is almost entirely concentrated below 2 Hz [60,61], whereas the oculomotor signals of interest, such as microsaccades and

**Table 2. Demographic and clinical characteristics of TD, ADHD, and drug-naïve ADHD participants.**

| | TD | ADHD | drug-naïve ADHD | *p*-values (TD vs ADHD) | *p*-values (TD vs drug-naïve ADHD) |
|---|---|---|---|---|---|
| Male/female | 8/12 | 8/8 | 4/7 | 0.73 | 1.00 |
| Age (year) | 37.0 ± 7.90 | 32.0 ± 8.29 | 28.5 ± 4.41 | 0.077 | **0.002** |
| Measurement start time (min) | 873.95 ± 136.03 | 861.50 ± 99.26 | 846.18 ± 91.12 | 0.761 | 0.550 |
| FIQ score | 104.8 ± 11.0 | 102.0 ± 14.3 | 105.7 ± 13.36 | 0.523 | 0.837 |
| VIQ score | 103.7 ± 10.6 | 103.4 ± 13.8 | 106.3 ± 12.31 | 0.949 | 0.530 |
| PIQ score | 105.1 ± 12.6 | 97.8 ± 16.4 | 100.7 ± 16.31 | 0.142 | 0.840 |
| ASRS Total score | 20.4 ± 11.0 | 42.9 ± 14.0 | 40.6 ± 11.11 | **<0.001** | **<0.001** |
| ASRS IN score | 12.5 ± 6.64 | 25.5 ± 6.86 | 24.7 ± 5.06 | **<0.001** | **<0.001** |
| ASRS Hyp/I score | 8.30 ± 5.42 | 17.6 ± 6.86 | 16.27 ± 7.76 | **<0.001** | **0.002** |

Physical characteristics in typical development (TD) and attention-deficit/hyperactivity disorder (ADHD) subjects. The group, which consists of ADHD subjects on medication and drug-naïve ADHD subjects, is represented by the ADHD group; the group which consists of only drug-naïve ADHD subjects, is represented by drug-naïve ADHD group. For group comparison sex ratio, the $\chi^2$-test was used. For the other group comparisons, including measurement start time, age, and IQ scores, a two-tailed *t*-test was used. The measurement start time indicates the recording start time expressed in minutes from midnight (0-1440). The *p*-values with *p* < 0.05 are represented by bold text. (FIQ, full-scale intellectual quotient; VIQ, verbal intellectual quotient; PIQ, performance intellectual quotient; ASRS, the Japanese version of the adult ADHD self-report scale; ASRS IN, ASRS inattentive; ASRS Hyp/I, ASRS hyperactivity/impulsivity).

drift, are prominent in the <10 Hz range [62]. Based on this, our primary analysis employed a 0.1–30 Hz band-pass filter to retain physiological ocular signals while suppressing very slow drifts and high-frequency noise. To rigorously test whether any residual head motion could have biased our results, we conducted a sensitivity analysis by re-running all analyses with more aggressive high-pass filters (2 Hz and 4 Hz). As detailed in our supplemental material (S2 Text), the results demonstrated that classifier performance remained highly stable across all filter settings for both the TD vs ADHD and TD vs drug-naïve ADHD comparisons. Notably, the model combining pupil diameter and a single feature, the temporal complexity of eye movement, which performed best in the primary analysis, consistently maintained its superior performance even after enhanced noise suppression. This outcome provides strong evidence that our findings are robust against head motion artifacts and that the features leveraged by our classifier are based on genuine physiological signals, rather than noise. Finally, surrogate-data analysis (Fig 1B) confirmed that the eye-movement time series exhibits non-linear dynamics, consistent with a neural control origin rather than stochastic motion noise. Nevertheless, we cannot completely rule out the possibility of residual head motion effects. Future studies would benefit from experimental setups that simultaneously record eye and head movements (e.g., using accelerometers or motion-tracking systems) to definitively resolve this issue.

Third, a potential limitation of our study is the exclusive use of mean pupil diameter as the sole pupillary metric. As noted in the Introduction—and as our own prior work has shown—pupil diameter variability (e.g., its temporal complexity) is an important indicator of ADHD pathophysiology [14,22]. Nevertheless, our primary aim was to determine the additional diagnostic value of a novel temporal complexity feature of eye movement when combined with a well-established pupillary marker. Therefore, to clearly isolate the contribution of this new feature—and to mitigate over fitting risks given our limited sample size—we selected mean pupil diameter. This decision was based on its utility as a foundational metric, supported by findings that adults with ADHD tend to exhibit larger baseline pupil diameters [14,21]. However, we acknowledge that variability-based metrics are likely to contain complementary information not captured by the mean alone [14]. Therefore, future studies should leverage larger datasets to develop more sophisticated classifiers that integrate the temporal complexity of both eye movements and pupil diameter, which may further improve diagnostic accuracy.

Fourth, a limitation is that neither pupil diameter nor the temporal complexity of eye movements showed a linear association with ADHD symptom severity on the ASRS (Fig 4). Several complementary explanations can be considered. First, statistical power was limited, as only 16 ADHD participants (11 drug-naïve) were available, and power analysis ($\alpha$ = 0.05, 80% power) suggests that approximately 60 cases would be required to confirm a modest correlation, such as the trend we observed ($r$ = –0.36) [63]. Second, the relationship between these biomarkers and symptom severity may be intrinsically non-linear. For example, the well-known inverted-U relationship between LC-NE arousal and cognitive performance, which is often indexed by pupil diameter [64,65], could obscure monotonic associations, and a similar non-linear profile may govern the temporal complexity of eye movements. Importantly, many diagnostic assays remain clinically valuable despite weak or absent correlations with disorder severity. For example, the SARS-CoV-2 cycle threshold (Ct) value and amyloid-PET imaging for Alzheimer's disease provide categorical diagnostic or prognostic value, even though their associations with clinical severity are modest or variable [66,67]. Similarly, our biomarkers may offer robust categorical discrimination to assist in early screening and differential diagnosis rather than tracking symptom severity. Future studies with larger, severity-stratified cohorts are needed to further elucidate these relationships and test for potential non-linear or subgroup-specific associations.

Finally, although this study revealed eye movement dynamics in adults with TD and those with ADHD, the relationship between the brain regions that control eye movements remains unclear. Therefore, a multimodal approach that simultaneously analyzes eye movements, EEG, and fMRI data is required. This approach is important for elucidating the relationship between eye-controlling brain regions and eye movement dynamics and for gaining a more comprehensive understanding of the neurobiological basis of ADHD. Moreover, in addition to ADHD, characteristic eye-movement disorders have been reported in cerebellar ataxia, Parkinson's disease, schizophrenia, and ASD [68,69]. Assessing eye movement dynamics in these disorders is important, and we plan to address these issues in future studies.

## Conclusions

In conclusion, this study reveals that the temporal complexity of eye movements is altered in ADHD. Additionally, machine learning has shown that combining these features with the pupil diameter index improves the accuracy of ADHD identification. The proposed evaluation method and findings highlight the usefulness of a multimodal approach, including pupil diameter and eye movements, for the diagnosis of ADHD and may contribute to improving diagnostic accuracy in future clinical practice.

## Methods

### Ethics statement

This study was conducted at the Medical Institute of Developmental Disabilities Research, Showa University, Japan, and was approved by the Ethics Committee of Showa University on December 25, 2015 (reference no. B-2015-025). A second approval, for the purpose of extending the study period, was granted on March 7, 2017, under the same reference number (Reference No. B-2015-025). All methods were performed in accordance with the Declaration of Helsinki. Participants were recruited between March 25, 2016, and September 28, 2018, at Seiwa Hospital, Tokyo, Japan. Before enrollment, each participant received a comprehensive explanation of the study and provided written informed consent. Since no minors were included in this study, consent from parents or guardians was not required. The authors confirm that they only accessed anonymized information during the data collection and analysis process and that no individually identifying information was available at any point.

### Participants

A total of 16 adult participants with ADHD (8 men, age: 32.0 ± 8.29 years) were recruited from outpatient consultations at Seiwa Hospital, Tokyo, Japan. They were diagnosed based on the criteria of the Diagnostic and Statistical Manual of Mental Disorders, Fifth Edition [1] through a semi-structured interview using the Assessment System for Individuals with ADHD [70]. The ADHD group consisted of 11 participants under drug-naïve conditions, and 5 participants were treated with methylphenidate (MHP; average dose, 45 mg/day) or atomoxetine (ATX; average dose, 80 mg/day). These 5 participants stopped taking ADHD medications on the experimental day. In this study, we defined 1-day medicine-free as being almost equal to drug-naïve, a paradigm which has been used in previous studies [14,21,22], because the average half-lives of methylphenidate and atomoxetine are 3.5 and 5 h, respectively. Participants with ADHD included 10 primarily inattentive (ADHD/I) and 6 combined inattentive/hyperactive (ADHD/C) individuals. To completely remove the influence of ADHD medications, a drug-naïve ADHD group consisting of 11 individuals (4 men, age: 28.5 ±

4.41 years) was established by excluding the drug-treated participants from the ADHD group.

We selected 20 adult participants with TD (8 men, age: $37.0 \pm 7.90$ years) to match the age, sex, and intelligence levels of the adult ADHD group [14,22]. Although the TD and drug-naïve ADHD group were matched for sex and intelligence, the drug-naïve ADHD group was younger. Detailed information on the subjects is provided in Table 2. To assess their intelligence levels, all participants were evaluated using the Wechsler Adult Intelligence Scale-Third Edition, revised Japanese edition (WAIS-Ⅲ), which includes measurements of intelligence quotient (IQ), verbal IQ, and performance IQ. To assess the participants' ADHD symptoms, they were subjected to the Japanese version of the Adult ADHD Self-Report Scale (ASRS) [8]. None of the TD participants displayed clinically significant levels of ADHD symptomatology as indexed by the ASRS. In all groups, we set the following exclusion criteria: current major depressive or manic-depressive episode, history of psychosis, Wechsler full-scale intelligence quotient < 80, history of head injury with loss of consciousness, sensory-motor handicap, or other neurological illnesses. All participants had normal or corrected-to-normal vision and hearing. Furthermore, all participants were of Japanese ethnicity, a population with predominantly dark brown iris color, which minimizes potential confounding factors related to this variable [71,72].

## Recording of eye movements and pupil diameter

To measure eye movements and pupil diameter, the participants sat in front of a monitor subtending $50.9 \times 28.6$ degrees of the visual angle at a 57 cm distance in a lit room. The participant's head was fixed with a chin rest. For 2 min, the participants fixed their gaze on a steady black cross ($0.87 \text{cd/m}^2$) subtending $0.5 \times 0.5$ degrees of the visual angle to obtain fixational eye movements and pupil diameter. The fixation cross was generated using Psychophysics Toolbox routines [73,74] for MATLAB (Version 2013b; MathWorks Ltd., http://www.mathworks.com/) and presented on a 23-inch LCD monitor ($1920 \times 1080$ pixels at 60 Hz) driven by a computer running Windows 7. During these measurements, the eye positions and pupil diameters of the participants were observed using a remote-type eye tracker (TX300; Tobii Technology, Stockholm, Sweden) with a sampling frequency of 300 Hz. On the day of the experiment, the participants did not ingest caffeine, nicotine, or any medication that could influence eye movement or pupil diameter. Additionally, to account for the potential effects of factors such as time of day and fatigue, the measurement start time for each participant was recorded. The times were converted to minutes from midnight for analysis, and no statistically significant differences were found between the groups (see Table 2 for details).

To analyze eye movements and pupil diameters, we performed the following preprocessing steps: First, the time-series data were divided into epochs of 10 seconds each. To eliminate the effect of blinking, data points within 10 ms before and after a blink were set as missing values [75]. Epochs with more than 50% of missing values were excluded from the analysis. The mean ($\pm$SD) proportion of excluded epochs was 12.1% ($\pm$27.9%) for the TD group, 20.3% ($\pm$29.2%) for the ADHD group, and 22.7% ($\pm$32.7%) for the drug-naïve ADHD group. There were no significant differences in these rates between the TD group and either the ADHD group ($p = 0.395$) or the drug-naïve ADHD group ($p = 0.348$), indicating that data quantity did not act as a confound. Subsequently, the remaining missing values within each epoch were filled using linear interpolation. The eye movement epochs were band-pass filtered between 0.1 and 30 Hz, whereas the pupil diameter epochs were low-pass filtered up to 5 Hz.

## Fuzzy entropy

FuzzyEn is a measure used to quantify the temporal complexity of time-series data. It improves the Approximate Entropy and Sample Entropy by being more consistent and less dependent on the data length [76–78]. The detailed steps for calculating FuzzyEn are as follows. First, for $N$ stochastic variables $\{x_1, x_2, \cdots x_N\}$ normalized by $z$-score, the following $m$-dimensional vectors were constructed:

$$X_i^m = \{x_i, x_{i+1}, \cdots, x_{i+m-1}\}, \qquad i = 1, 2, \cdots N - (m - 1). \tag{1}$$

The distance between each $X_i^m$ and $X_j^m$ is defined as $d_{ij}^m = d[X_i^m, X_j^m]$ given by the Chebyshev distance. Specifically, distance is expressed by the following equation:

$$d[X_i^m, X_j^m] = \max_{0 \leq k \leq m-1} \{|x_{i+k} - x_{j+k}|\}, \qquad i, j = 1, 2, \cdots N - (m - 1), i \neq j. \tag{2}$$

Given FuzzyEn with power $n$ and tolerance $r$, the similarity degree $d_{ij}^m$ is calculated using a fuzzy membership function, which is an exponential function $\mu(d_{ij}^m, n, r) = \exp\left(-\frac{(d_{ij}^m)^n}{r}\right)$. Function $\phi^m$ is then defined as

$$\phi^m(x, n, r) = \frac{1}{N - m} \sum_{i=1}^{N-m} \frac{1}{N - m - 1} \sum_{j=1, i \neq j}^{N-m} \exp\left(-\frac{(d_{ij}^m)^n}{r}\right). \tag{3}$$

Finally, fuzzy FuzzyEn of the signal is defined as a negative natural logarithm of the ratio of $\phi^m$ to $\phi^{m+1}$ (computed following the same procedure as for the embedding dimension $m + 1$)

$$\text{FuzzyEn}(x, m, n, r) = -\ln \frac{\phi^{m+1}}{\phi^m}. \tag{4}$$

## Multi-scale entropy (MSE) analysis by coarse-graining

MSE analysis was used to assess the temporal scale dependence of the temporal complexity of the eye movements. First, a coarse-graining process is applied to an observed time series $\{y_1, y_2, \cdots y_M\}$ where $M$ is the length of the signal. Each element of the coarse-grained time series $\{x_1, x_2, \cdots x_N\}$ for MSE is defined as follows:

$$x_j = \frac{1}{\tau} \sum_{j=(j-1)\tau+1}^{j\tau} y_j, \qquad 1 \leq j \leq \left\lfloor \frac{M}{\tau} \right\rfloor = N. \tag{5}$$

Here, $\tau(\tau = 1, 2, \cdots)$ represents temporal scale. For the MSE analysis, FuzzyEn was derived using Eq (4) for the coarse-grained time series $\{x_1, x_2, \cdots x_N\}$ at each temporal scale $\tau$. In the study, we set $m = 2$, $n = 2$, and $r = 0.2$ [76,78,79]. The FuzzyEn values at each temporal scale $\tau$ ($\tau = 1, 2, \cdots, 30$) corresponding to 0.003–0.1 s were averaged for each epoch.

## Surrogate data analysis

To test whether the observed temporal complexity in eye movements indicates the presence of nonlinear dynamics, rather than originating from mere stochastic noise, we performed a surrogate data analysis using the iterative amplitude-adjusted Fourier transform (IAAFT) method [80]. The null hypothesis ($H_0$) for this test is that the time series is generated by a

stationary linear stochastic process that has undergone a static and monotonic, nonlinear transformation. IAAFT surrogates preserve the original signal's power spectrum (i.e., linear autocorrelations) and amplitude distribution (i.e., the frequency distribution of the sample values) while randomizing Fourier phases to destroy any nonlinear temporal structure; this provides a more conservative control than simple random shuffling, which eliminates all temporal structure [81]. Therefore, rejecting the null hypothesis by finding a significant difference between the original and surrogate data provides strong evidence that the time series contains nonlinear dynamics that cannot be explained by a linear stochastic process alone. For each original eye movement recording, ten IAAFT surrogate datasets were generated using different random seeds and an iteration number of 100. The FuzzyEn values from the original data were then compared with the average FuzzyEn values from the corresponding surrogates.

### Analysis of mean pupil diameter

To quantify each participant's overall pupil size, we calculated the mean pupil diameter, which is defined as the temporal average of the pupil diameter across all valid epochs of the complete fixation period. These values were then averaged between the right and left eyes.

### Statistical analysis

The FuzzyEn values at each temporal scale were found to have a skewed distribution and were therefore log-transformed to approximate a normal distribution (see S1 Fig for representative probability-density plots of the scale-averaged features before and after transformation) [36,37]. For the surrogate data analysis, paired $t$-tests were performed to compare the FuzzyEn values of the original eye movement time series at each temporal scale $\tau$ ($\tau = 1, 2, \cdots, 30$) with the values of the IAAFT surrogate time series in each group (TD, ADHD, and drug-naïve ADHD). Benjamini-Hochberg false discovery rate (FDR) correction was applied to the $t$-scores for multiple comparisons ($q < 0.05$ and $q < 0.01$) across the 30 $p$-values.

For MSE analysis, ANCOVA was performed to test for group differences in FuzzyEn values of the original eye movement time series. The ANCOVA included group (TD vs ADHD or drug-naïve ADHD) as a between-subjects factor, temporal scale ($\tau$: 1–30) as a within-subjects factor, and age as a covariate. Greenhouse-Geisser adjustments were applied to the degrees of freedom. A two-tailed $\alpha = 0.05$ was used. Post hoc $t$-tests were then performed to assess the group effects or group × temporal-scale interactions. FDR correction was applied to the $t$-scores for multiple comparisons ($q < 0.05$ and $q < 0.01$) across 30 $p$-values.

For the analysis of mean pupil diameters, one-way ANCOVA was performed to test for group differences. ANCOVA included group (TD vs ADHD or drug-naïve ADHD) as a between-subjects factor and age as a covariate. Greenhouse-Geisser adjustments were applied to the degrees of freedom. A two-tailed $\alpha = 0.05$ was used. Post hoc $t$-tests were used to assess the significant main effect of the groups. In $t$-tests, a two-tailed $\alpha$ level of 0.05 was considered significant.

Using Pearson's correlation coefficients, we evaluated how the severity of ADHD symptoms, measured using ASRS scores, was related to Hor FuzzyEn, Vert FuzzyEn, and pupil size. To this end, we used the total ASRS and ASRS subscores, namely the ASRS hyperactivity/impulsivity (ASRS Hyp/I) and ASRS inattentive (ASRS IN) scores. All statistical analyses were performed using IBM SPSS Statistics (v29.0.2.0 (20)) or MATLAB (ver R2023b).

## Machine learning for classification

To discriminate between individuals with ADHD and TD, we employed an L1-regularized (Lasso) logistic regression model as the primary classifier [38]. The Lasso penalty shrinks the coefficients of uninformative predictors toward zero, thereby mitigating overfitting [38]. This strategy is well-suited to datasets like ours, which have a modest sample size (Table 2) and correlated features (S1 Table). The model utilized three predictive features: mean pupil diameter (Pupil Size), average Fuzzy Entropy of horizontal eye movements (Hor FuzzyEn), and average Fuzzy Entropy of vertical eye movements (Vert FuzzyEn). For comparison, standard logistic regression and Firth's logistic regression were also evaluated as supplemental models (see S3 Table) [57].

Several preprocessing steps were performed prior to model training. The features were standardized using a z-score transformation. To prevent data leakage, the mean and standard deviation for this transformation were calculated exclusively from the training data within each fold of the cross-validation process and were subsequently applied to both the training and test sets. Furthermore, to address the moderate class imbalance observed between the groups (see Table 2), a weighted learning approach was implemented [56]. This method assigned a weight to each sample that was inversely proportional to its class frequency, thereby preventing the classifier from being biased towards the majority class.

For robust model development and unbiased performance evaluation, a nested cross-validation framework was adopted [82]. The outer loop consisted of a Leave-One-Out Cross-Validation (LOOCV) to provide a reliable estimate of the model's generalization performance [83]. Within each training fold of this outer loop, an inner stratified 5-fold cross-validation was used for the hyperparameter tuning. The optimal regularization parameter, Lambda, was determined through Bayesian optimization [84], which systematically searched for the value that maximized the classification performance on the inner-loop validation sets.

The predictive performance of the final model was comprehensively evaluated based on the predictions from the LOOCV [83]. The overall discrimination ability was assessed using both the Receiver Operating Characteristic (ROC) and precision-recall (PR) curves, along with their respective area under the curve metrics (AUC-ROC and AUC-PR). The use of the PR curve and its AUC is particularly informative for evaluating performance of datasets with class imbalance [39]. The optimal classification threshold was defined as the point on the PR curve that maximized the F1-score. Using this threshold, the final performance was reported using confusion matrices, detailing the sensitivity and specificity. The outcomes of this evaluation framework are presented in Fig 3, S2 Table, and S3 Table. All machine learning analyses were performed in MATLAB (ver R2023b) using the Statistics and Machine Learning Toolbox.

## Supporting information

**S1 Table. Correlation matrix of predictive features.**
(PDF)

**S2 Table. Performance of the Lasso logistic regression model across all feature combinations.**
(PDF)

**S3 Table. Performance comparison across different classifiers.**
(PDF)

**S1 Text. Analysis of Gaze Distribution.**
(PDF)

**S2 Text. Sensitivity Analysis.**
(PDF)

**S1 Fig. Distributions of FuzzyEn features before and after log-transformation.**
(PDF)

**S1 Data. Detailed participant-level data (demographics, clinical scores, MSE measures, mean pupil diameter, etc.).**
(CSV)

## Author contributions

**Conceptualization:** Ayumu Ueno, Sou Nobukawa, Aya Shirama.

**Data curation:** Aya Shirama.

**Formal analysis:** Ayumu Ueno, Sou Nobukawa.

**Funding acquisition:** Sou Nobukawa, Aya Shirama.

**Investigation:** Ayumu Ueno, Sou Nobukawa, Aya Shirama.

**Methodology:** Ayumu Ueno, Sou Nobukawa.

**Project administration:** Sou Nobukawa, Aya Shirama.

**Resources:** Aya Shirama.

**Software:** Ayumu Ueno.

**Supervision:** Sou Nobukawa.

**Validation:** Ayumu Ueno, Sou Nobukawa, Aya Shirama.

**Visualization:** Ayumu Ueno.

**Writing – original draft:** Ayumu Ueno, Sou Nobukawa.

**Writing – review & editing:** Ayumu Ueno, Sou Nobukawa, Aya Shirama.

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
