## [Decision Letter · Decision Letter 0]

23 Jun 2025

PMEN-D-25-00061

Multimodal biomarker based on eye movement complexity and pupil diameter in attention-deficit/hyperactivity disorder

PLOS Mental Health

Dear Dr. Nobukawa,

Thank you for submitting your manuscript to PLOS Mental Health. After careful consideration, we feel that it has merit but does not fully meet PLOS Mental Health’s publication criteria as it currently stands. Therefore, we invite you to submit a revised version of the manuscript that addresses the points raised during the review process.

We look forward to receiving your revised manuscript.

Kind regards,

Zhiyi Chen

Academic Editor

PLOS Mental Health

Journal Requirements:

Additional Editor Comments (if provided):

Thank you for the patience in waiting this decision. Though the Reviewer #2 recommend to accept this manuscript in the current form, I do reckon that the Reviewer #1 raised substantial solid concerns warranting an overhaul. I thus recommend a major revision before publication.

Reviewers' comments:

Reviewer's Responses to Questions

**Comments to the Author**

1. Does this manuscript meet PLOS Mental Health’s publication criteria? Is the manuscript technically sound, and do the data support the conclusions? The manuscript must describe methodologically and ethically rigorous research with conclusions that are appropriately drawn based on the data presented.

Reviewer #1: Yes

Reviewer #2: Yes

2. Has the statistical analysis been performed appropriately and rigorously?

Reviewer #1: N/A

Reviewer #2: Yes

3. Have the authors made all data underlying the findings in their manuscript fully available (please refer to the Data Availability Statement at the start of the manuscript PDF file)?

Reviewer #1: Yes

Reviewer #2: Yes

4. Is the manuscript presented in an intelligible fashion and written in standard English?

Reviewer #1: Yes

Reviewer #2: Yes

5. Review Comments to the Author

Reviewer #1: Peer Review

Multimodal biomarker based on eye movement complexity and pupil diameter in ADHD

I am not an expert on oculomotorics or ADHD, but rather neurological biomarkers for other psychopathologies, so my review mostly focusses on conceptual clarity and the statistical methods. Nonetheless, the argument for the exploration of the role of eye movements and pupil size in ADHD seems plausible and well-contended by literature. In the following, I delineate major and minor concerns that arose in the process of reviewing this paper.

Major concerns:

- The nature of the recorded (/alleged) eye movements

o Howcantheauthorsensurethatthemeasuresfedintotheanalysesareindeed eye movements, and not head motion artifacts? Especially with ADHD, which is associated with the inability to stay still.

o Evenwithachinrest,Idoubtthelackofmotion-inducednoiseinthedata

o Toaddressthis,eitherathoroughpreprocessingremovingartifactsisnecessary, or at least the addition of a heatmap of the fixations to the paper. This additional figure could represent the single observations, or it could be the mean of each subgroup; the important thing would be to assess how big the eye movements are (because realistically, micro-saccades are associated with ADHD, not big eye

movements)

- AUC may confound interpretation

o AUCcurvescanoftenseemverygood,butobscurethetrueclassificatoryprowess of models

o For example, a 2x2 confusion matrix showing the exact classifications could clarify the aptitude of the predictors and the models in general

o Istronglysuggesttheadditionofaprecision-recallcurvetocomplementtheAUC curves

- Logistic models

o Logisticregressionsoftenoverfitmodelswithsuchsmallsamplesizes,evenwith

cross-validation. Was any alternative method considered?

o It is good that ROC curves and cross-validation accuracy were reported, but it

should be acknowledged in the limitations paragraph of the discussion.

o Additionally, the unbalanced group sizes can skew results and should also be

acknowledged.

o Althoughthesmallsamplesizewasaddressed,theimplicationsofthisweren’t

and can result in misinterpretation on the generalizability and veracity of the findings.

Minor concerns:

- The authors should discuss or hypothesize why the measures do not correlate with ADHD

severity, given that the measures are supposed to serve as an ADHD biomarker. This finding is counter-intuitive and should thus be addressed. Discuss what the utility of a biomarker is that does not diVer between disorder severity?

- Histograms of the main measures/predictors (before log-transformations) would be helpful to see if the distributions overlap

- Clarity of concepts

o Someconceptsanddisorderscouldhavebeenbrieflydescribed,sincesomeare

not commonly known of, e.g., “dysmetria”. A short sentence like “with and without dysmetria, a neurological disorder aVecting fine motor skills....” would already suVice.

o Similarly, defining the meaning of eye movement complexity in the beginning would facilitate understanding, as the term is used for the general concept of eye

movement, and for the time-adapted operationalization in this paper, i.e., temporal complexity. It is often diVuse how eye movement complexity is operationalized through MSE as FuzzyEn, and when the general term is meant, or the temporal concept.

o Similarly, “baseline pupil diameter” suggests that pupil diameter was only assessed at the beginning of the measurement, instead of the time-averaged measure that it is. Perhaps rephrasing “baseline pupil diameter” to “mean pupil diameter during the complete fixation period” clarifies this.

- Measurement conditions

o Theexperimentalconditionsarenotfullydescribedinmyopinion.

o The utilization of mean pupil diameter seems in agreeance with literature, but

diameter is heavily influenced by other factors. For example, was every participant measured at the same time of the day, since fatigue plays a role in pupil size? Even eye color can play a role, since, e.g., with the same amount of light, darker eyes tend to have less dilated pupils than lighter colored eyes. Was any of this controlled for? If so, it should be described in the section on measurement.

- Main measures: Pupil size

o Was it considered to assess pupil size variability, not just pupil size mean?

Variability may have more classification power than just utilizing the mean, indicated by the statement in lines 44 to 46 (“the temporal complexity of pupil diameter is important for capturing the pathophysiology of ADHD, and its relationship with the LC-NE system has been reported”).

o Perhapsananalysisincludingpupilvariabilityorsomekindoftimevariancedata on pupil size can be added.

- Multicollinearity

o HowwasthethresholdforPearson’srtoassessmulticollinearityestablished?Are

there guidelines for this?

o Whyweren’tothermulticollinearitymeasuresusedinstead,e.g.,VIForVDPs?

o Multicollinearitycould’vebeentestedinthemodelsthemselves,withouthaving

to remove multicollinear eVects a priori

- Was the epoch rejection/inclusion rate similar across the three subgroups, since noisy

epochs were removed? This could also influence results.

- Interpretability of the MSE results

o ComplexitydiVerencesmightindeedreflectneurophysiologicalalterationsorbe mere noise. A further control analysis, for example using completely shuVled time series, could confirm the findings and substantiate that entropy diVerences are not driven by measurement artifacts.

My recommendation is to accept the paper after amending it with some supplementary control analyses as indicated above. If the additional analyses change the interpretation of the results, a broader review & rewrite of the paper by the authors will be necessary.

Reviewer #2: This article presents a novel and promising approach to identifying ADHD using a multimodal biomarker that integrates eye movement complexity and pupil diameter. The methodology is technically sound, with appropriate use of signal analysis and statistical validation. The study highlights the potential of non-invasive, objective tools to enhance diagnostic accuracy in ADHD, which is traditionally reliant on subjective assessments. However, while the findings are compelling, the sample size is limited, and external validation is needed. Overall, the research contributes meaningfully to the field of neuropsychiatric diagnostics and paves the way for future clinical applications

6. PLOS authors have the option to publish the peer review history of their article (what does this mean?). If published, this will include your full peer review and any attached files.

**Do you want your identity to be public for this peer review?** For information about this choice, including consent withdrawal, please see our Privacy Policy.

Reviewer #1: No

Reviewer #2: No

---

## [Decision Letter · Decision Letter 1]

18 Sep 2025

Multimodal biomarker based on temporal complexity of eye movements and pupil diameter in attention-deficit/hyperactivity disorder

PMEN-D-25-00061R1

Dear Professor Nobukawa,

We are pleased to inform you that your manuscript 'Multimodal biomarker based on temporal complexity of eye movements and pupil diameter in attention-deficit/hyperactivity disorder' has been provisionally accepted for publication in PLOS Mental Health.

Best regards,

Zhiyi Chen

Academic Editor

PLOS Mental Health

All the concerns have been addressed well. I am pleased to accept this manuscript in the current form.

Reviewer Comments (if any, and for reference):

Reviewer's Responses to Questions

**Comments to the Author**

1. If the authors have adequately addressed your comments raised in a previous round of review and you feel that this manuscript is now acceptable for publication, you may indicate that here to bypass the “Comments to the Author” section, enter your conflict of interest statement in the “Confidential to Editor” section, and submit your "Accept" recommendation.

Reviewer #1: All comments have been addressed

2. Does this manuscript meet PLOS Mental Health’s publication criteria? Is the manuscript technically sound, and do the data support the conclusions? The manuscript must describe methodologically and ethically rigorous research with conclusions that are appropriately drawn based on the data presented.

Reviewer #1: (No Response)

3. Has the statistical analysis been performed appropriately and rigorously?

Reviewer #1: (No Response)

4. Have the authors made all data underlying the findings in their manuscript fully available (please refer to the Data Availability Statement at the start of the manuscript PDF file)?

Reviewer #1: (No Response)

5. Is the manuscript presented in an intelligible fashion and written in standard English?

Reviewer #1: (No Response)

6. Review Comments to the Author

Reviewer #1: (No Response)

7. PLOS authors have the option to publish the peer review history of their article (what does this mean?). If published, this will include your full peer review and any attached files.

**Do you want your identity to be public for this peer review?** For information about this choice, including consent withdrawal, please see our Privacy Policy.

Reviewer #1: No
